# Development of LAT1-Selective Nuclear Medicine Therapeutics Using Astatine-211

**DOI:** 10.3390/ijms252212386

**Published:** 2024-11-18

**Authors:** Kazuko Kaneda-Nakashima, Yoshifumi Shirakami, Kentaro Hisada, Sifan Feng, Yuichiro Kadonaga, Kazuhiro Ooe, Tadashi Watabe, Yoshiyuki Manabe, Atsushi Shimoyama, Masashi Murakami, Atsushi Toyoshima, Hiromitsu Haba, Yoshikatsu Kanai, Koichi Fukase

**Affiliations:** 1Radiation Biological Chemistry, MS-CORE, FRC, Graduate School of Science, Osaka University, 1-1 Machikaneyama-cho, Toyonaka, Osaka 560-0043, Japan; 2Institute for Radiation Sciences, Osaka University, 2-4 Yamadaoka, Suita, Osaka 565-0871, Japan; 3Department of Radiology, Graduate School of Medicine, Osaka University, 2-2 Yamadaoka, Suita, Osaka 565-0871, Japan; 4Natural Product Chemistry, Department of Chemistry, Graduate School of Science, Osaka University, 1-1 Machikaneyama-cho, Toyonaka, Osaka 560-0043, Japan; 5Nishina Center, RIKEN, 2-1 Hirosawa, Wako, Saitama 351-0198, Japan; 6Premium Research Institute for Human Metaverse Medicine, Osaka University, 2-2 Yamadaoka, Suita, Osaka 565-0871, Japan

**Keywords:** astatine-211, LAT1, amino acid, cancer therapy, nuclear medicine

## Abstract

We investigated nuclear medicine therapeutics targeting the L-type amino acid transporter 1 (LAT1). We previously reported that a nuclear medicine therapeutic drug using astatine 211 (^211^At), an alpha-emitting nuclide that can be produced in an accelerator and targets LAT1 as a molecular target, is effective. The seed compound was 3-[^211^At] Astato-α-methyl-L-tyrosine (^211^At-AAMT-OH-L). We used a unique labeling method. By changing the OH group of phenol to a methyl group, retention was successfully increased. It was also found that the amount of the L-isomer taken up by the D-isomer and L-isomer was clearly higher, and the L-isomer was superior as a therapeutic drug. Compounds in which the methyl group was replaced with an ethyl or propyl group were also examined, but their retention did not increase significantly. In fact, we observed increased non-specific accumulation and dynamics, suggesting that labeling may be off. In addition, ^211^At-AAMT-O-Me-L, which has a simple structure, was clearly superior in terms of uptake speed for several candidate compounds. As a result, we were able to develop a compound that can be easily labeled, has high specific radioactivity, is stable, and has a strong therapeutic effect.

## 1. Introduction

Currently, we are developing an alpha-nuclear medicinal therapeutic agent using astatine-211 (^211^At), an alpha-emitting nuclide. ^211^At, which we use, is a nuclide that can be produced using an accelerator. Therefore, ^211^At has attracted particular attention in Japan as a labeling nuclide for alpha-emitting nuclear medicine. Compared with Actinium-225 (half-life: 10 days) and Radium-223 (half-life: 11.2 days), which are also attracting attention, ^211^At has a shorter half-life (7.2 h), which results in fewer side effects and a shorter hospitalization time for the patient. Numerous review articles have been published on the application of ^211^At in nuclear medicine drugs [1,2,3,4,5,6,7,8]. Therefore, the usefulness of ^211^At is widely known. Furthermore, ^211^At is a halogen nuclide that is relatively easy to label, and techniques for labeling various compounds are being investigated [9]. The use of compounds with short half-lives, such as astatine-211, as nuclear medicine therapeutics requires rapid labeling. Transport time must be considered in addition to labeling time if nuclides cannot be made in one’s own tissues. Osaka University is also developing an accelerator that is easy to install for convenient production [10].

To date, we have reported a nuclear medicine therapeutic drug that molecularly targets LAT1 [11,12], a cancer cell-type amino acid transporter. LAT1 was identified by Kanai et al. [13] and has been proven to be a useful molecular target for Positron Emission Tomography (PET) imaging because of its cancer specificity [14,15,16,17]. It is also an excellent molecular target that has been considered a potential target for Boron Neutron Capture Therapy (BNCT) therapy [18,19,20,21,22]. This research was conducted with the aim of developing a compound with a higher labeling efficiency and a higher accumulation rate in cancer tissues. First, we investigated the increase in lipid solubility to increase its ability to accumulate in tumor tissues. The selectivity of LAT1 is proportional to the size of its side chain [23], and its cellular uptake is inversely proportional. Next, we considered the specific radioactivity. This is because the study of efficient labels will not only be useful for future pharmaceuticals, but also enable the efficient use of nuclides. The alpha-methyl group is crucial for maintaining the selectivity of LAT1. In addition, the seed compound had some stability problems owing to the presence of a hydroxy group near the ^211^At labeling site. Therefore, we aimed to improve the stability and retention of the compound without altering its selectivity by modifying its side chain. Appendix A shows the compounds used in this study. Previously, we used the mercury method for labeling; however, in 2021, we developed a revolutionary method called the “*Shirakami reaction*”. This labeling method, discovered by the co-author Pf. Shirakami, is a method that replaces ^211^At with high efficiency by introducing boronic acid or pinacolborane into the site to be labeled and is safe as it does not require any special reagents [24].

There are two methods to impart LAT1 selectivity to compounds. One method is to introduce a bulky hydrophobic substituent at the 4-position of the aromatic ring in the side chain and the other is to add a methyl group to the α-carbon. In the former case, although it is possible to obtain high affinity, the transport activity decreases, and its properties as a blocker come to the fore, as with melphalan. In the latter case, it becomes LAT1 selective while retaining its transport activity. Based on this characteristic, we aimed to improve fat solubility by slightly elongating the side chains. First, the methoxy form of the seed compound was synthesized as a candidate compound. To optimize our compounds, we first compared them with the seed compound ^211^At-AAMT (^211^At-AAMT-OH-L) and candidate compounds. We then evaluated the optical isomers and compared the effects of the side-chain modifications to determine the optimal compound.

## 2. Results

### 2.1. Synthesis of ^211^At Labeled Chemicals

All precursors were labeled by “*Shirakami Reaction*” and purified using an HLB column (Waters Corp, Waltham, MA, USA). All structures are shown in Appendix A. After purification, the solution was supplemented with 1% (*v*/*w*) of ascorbic acid. The retention time of (S)-2-amino-3-(3-(astato-^211^At)-4-methoxyphenyl)-2-methylpropanic acid (^211^At-AAMT-OMe-L) determined using High Performance Liquid Chromatography (HPLC) (concentration: 1 MBq/mL, Flow: 1 mL/min; absorbance = 226 nm) was 12.91 min. A COSMOSIL Packed 5C_18_-MS-II (4.6 mm × 150 mm) Column was used at 40 °C. Mobile phase A was 0.1% formic acid and B was acetonitrile (10–60%, 20 min). HPLC analysis revealed that it was a single-peak compound (Figure 1A). The stability of ^211^At-AAMT-OMe-L in its unpurified state was analyzed by thin layer chromatography (TLC). BAS IP MS 2025E (GE Healthcare Technologies Inc., Chicago, IL, USA) was used for imaging. Image data were collected using a Tyhoon^TM^ FLA-7000 (Cytiva, GE Healthcare Technologies Inc.). The eluent was a 1:2 mixture of water and acetonitrile, and the Rf ratio was 0.80. The compound showed almost no degradation after 24 h and 48 h (Figure 1B). Compounds labeled using the mercury method were unstable without the addition of an antioxidant (ascorbic acid) [12]; on the other hand, the data for ^211^At-AAMT-OMe-L shown in Figure 1B show that the compound made by the *Shirakami reaction* is stable. We also labeled ^211^At-AAMT-O-Me-D, ^211^At-AAMT-O-Et-L, and ^211^At-AAMT-O-Pr-L according to the same method. The labeling results are shown in Appendix A. Optimization of the labeling conditions revealed that ^211^At-AAMT-OMe-L could be labeled using the smallest amount of raw material. Because optical isomers were only examined for ^211^At-AAMT-OMe-L and the remaining labeled products were all L-isomers, the DL notation was omitted when optical isomers were not compared. ^211^At-AAMT-O-Me-L in the crude product solution was detected at Rf = 0.80, yielding a radiochemical yield (RCY) of 81.3%. The radiochemical yield and purity of the purified product were 81% and 98%, respectively. ^211^At-AAMT-O-Me-D in the crude product solution was detected at Rf = 0.80, yielding a radiochemical yield (RCY) of 79.6%. The radiochemical yield and purity of the purified product were 79.6% and 99.9%, respectively. ^211^At-AAMT-O-Et-L in the crude product solution was detected at Rf = 0.80, yielding a radiochemical yield (RCY) of 85.5%. The radiochemical yield and purity of the purified product were 75.9% and 99.27%, respectively. ^211^At-AAMT-O-Pr-L in the crude product solution was detected at Rf = 0.80, yielding a radiochemical yield (RCY) of 78.0%. The radiochemical yield and radiochemical purity of the purified product were 78.0% and 99.52%, respectively.

### 2.2. Uptake Comparison Between Candidate Compounds

Each cell line was seeded in 24 wells at 1 × 10^5^ cells/mL and allowed to become sub-confluent on the day of the experiment. Cells were washed with PBS (-) (Fujifilm Wako Pure Chemical, Osaka, Japan) prior to the addition of the compound, which was then replaced with uptake buffer [25]. After treating the cells with the same dose, they were collected at predetermined time points. The cell dose was measured using a γ-counter (Wizard^2^ 2480; PerkinElmer, Waltham, MA, USA), and the protein content of the sample was determined. Protein content was measured using the BCA method, and the dose was corrected using the protein content (Protein Assay BCA kit, Fujifilm Wako Pure Chemical).

Comparison of uptake among the ^211^At labeled compounds confirmed that ^211^At-AAMT-O-Me-L showed the highest increase in uptake in accordance with a LAT1 expression (doxycycline concentration)-dependent manner (Figure 2A). A comparison of the time-dependent uptake of the compounds in PANC-1 cells also revealed that ^211^At-AAMT-OMe-L showed the highest uptake. Only ^211^At-AAMT-OMe-L showed a significantly higher uptake than the seed compound ^211^At-AAMT-OH-L (Figure 2B).

Differences in optical isomers were examined using the LAT1 overexpression system in HEK293 cells, and there were no significant differences in specific uptake, suggesting comparable performance in in vitro experiments (Figure 3A). We also confirmed a study using cancer cells and found that in cancer cells, the L-isomer was taken up more and the D-isomer was taken up less (Figure 3B).

### 2.3. Distribution Differences Between Candidate Compounds In Vivo

Table 1 shows a comparison between the seed compound ^211^At-AAMT-OH-L and the candidate compound ^211^At-AAMT-O-Me-L. Both compounds were L-isomers. The major difference in the comparison of distribution is the reduced accumulation in the bone, including the bone marrow, cecum, and pancreas. Conversely, the accumulation was higher in the salivary glands, lungs, spleen, and stomach. By 24 h later, the residual amount in the tumor was not large in both compounds, but it was confirmed that the residual amount in the tumor was higher with ^211^At-AAMT O-Me-L than ^211^At-AAMT-OH-L.

First, the distributions of the D- and L-isomers were compared between ICR and normal animals. Table 2 shows the data at 1 h and 3 h after injection. Both compounds accumulated in the thyroid gland, salivary glands, and stomach at the 1 h point, but the thyroid and salivary gland accumulation was lower at 3 h. The accumulation in the stomach remained high, but the dose in the small intestine was higher. The largest difference in the distribution of the optical isomers was in urinary excretion: at the 1 h point, more of the L-isomer was excreted, but at the 3 h point, more of the D-isomer was excreted. In ICR mice as normal animals, the accumulation of both compounds in the thyroid and stomach is higher. Accumulation in the thyroid was high at the 1 h point (D-isomer in thyroid: 8.06 ± 3.56; L-isomer in thyroid: 13.12 ± 1.72) but was suppressed to a low level at the 3 h point (D-isomer in thyroid: 0.42 ± 0.20; L-isomer in thyroid: 0.60 ± 0.16). Comparing the blood doses, the D-isomer dose increased faster in the blood and then decreased because of urinary excretion. The L-isomer dose was almost unchanged in the blood at the 1 and 3 h points, suggesting that retention was higher for the L-isomer. Accumulation in the thyroid and stomach was low in carrier animals. The blood dose was higher for the L-isomer at both the 1 and 3 h time points, and the %ID/g in the tumor tissue was clearly higher for the L-isomer.

The distribution data of the optical isomers in the PANC-1 tumor-bearing model are listed in Table 3. Comparing the D- and L-isomers, the L-isomer accumulation was higher at both 1 and 3 h. Although tumor accumulation is affected by the number of blood vessels infiltrating the tumor tissues, LAT1 is expressed on cancer cells and the number of formed stromal cells. The L-isomer showed higher accumulation in tumors than under the same tumor conditions. The amount of excretion was also higher for the D-isomer in the tumor-bearing model.

We changed the side chain in the methyl group (^211^At-AAMT-OMe-L) to an ethyl group (^211^At-AAMT-OEt-L) with the aim of increasing accumulation in tumors. The distribution data in ICR mice or nude mice (PANC-1 tumor-bearing model) were shown in Table 4. In ICR mice, the accumulation of ^211^At-AAMT-O-Et-L in non-specific organs was very high, and side effects were feared (Table 4). In nude mice, tumor accumulation and non-specific accumulation in non-tumor organs were also higher (Table 4). ^211^At-AAMT-O-Pr-L was constructed for the same purpose as that of ^211^At-AAMT-O-Et-L. ^211^At-AAMT-O-Pr-L was investigated in the same manner as ^211^At-AAMT-O-Et-L. Increased accumulation was observed in the thyroid gland, salivary glands, and stomach. In the PANC-1 tumor-bearing model, increased accumulation was observed in the pancreas and spleen (Table 5).

### 2.4. Excretion Differences Between Candidate Compounds In Vivo

Urinary excretion levels were also examined (Appendix A). Comparing the D-isomer and L isomer of ^211^At-AAMT-O-Me from ICR mice, the D-isomer; 24.84 ± 13.03% ID, L-isomer; 33.69 ± 1.26% ID at 1 h after injection. D-isomer; 71.36 ± 4.67% ID, L-isomer; 23.99 ± 2.40% ID at 3 h after injection. From PANC-1 tumor bearing mice, D-isomer; 35.24 ± 8.70% ID, L-isomer; 20.40 ± 0.76% ID at 1 h after injection. D-isomer; 47.38 ± 3.19% ID, L-isomer; 3.23 ± 0.67% ID at 3 h after injection. In tumor-bearing models, the amount of excretion was smaller than that in normal mice. We have already reported that, in the presence of target cancer tissue for administered drugs, retention in the cancer tissue results in reduced excretion compared with normal animals. These results were confirmed by our previous study [26]. The same was also observed for feces, but very little was excreted in the feces. The results were as follows: from ICR mice, D-isomer; 0.06 ± 0.03% ID, L-isomer; 0.004 ± 0.003% ID at 1 h after injection. D-isomer; 0.02 ± 0.01% ID, L-isomer; 0.02 ± 0.01% ID at 3 h after injection. From PANC-1 tumor bearing mice, D-isomer; 0.25 ± 0.12% ID, L-isomer; 0.002 ± 0.001% ID at 1 h after injection. D-isomer; 0.75 ± 0.46% ID; L-isomer; 0.07 ± 0.03% ID at 3 h after injection.

### 2.5. Anti-Tumor Effect in Xenograft Model

See Figure 4 for a comparison of the anti-tumor effects of the compounds in the PANC-1 tumor model (Figure 4). Mice in the control group were euthanized when the tumor size exceeded 10% of their body weight. Tumor growth was suppressed after 7 to 10 days. Compounds with higher accumulation in tumor tissues tended to have greater therapeutic efficacy, but there was no significant difference between the compounds.

No significant differences were observed between the compounds in the PANC-1 tumor bearing model, and all compounds showed anti-tumor effects.

## 3. Discussion

To compare the time of accumulation in tumors, ^211^At-AAMT-OH-L and ^211^At-AAMT-O-Me-L were compared in terms of their accumulation in tissues after 24 h (Table 1). Labeling was performed using the “*Shirakami reaction*”. Both compounds were eliminated from the body after 24 h, but ^211^At-AAMT-O-Me-L remained 2.6 times more accumulated in the tumors than ^211^At-AAMT-OH-L. In addition, the blood intensity was 22 times higher. In contrast, ^211^At-AAMT-OH-L remained in the thyroid (5.2-fold), pancreas (4.1-fold), cecum (3.2-fold), and femur, including the bone marrow (5.7-fold), and was thought to induce side effects more easily than ^211^At-AAMT-O-Me-L (Table 1). Because ^211^At-AAMT-O-Me-L was found to be a superior labeling compound, we next compared its optical isomers. ICR mice were used as normal animals, and comparisons were made 1 and 3 h after administration. The results showed that 1 h after administration, astatine itself is known to have a higher accumulation rate [27] in the thyroid gland, stomach, and urine, and the L-isomer was found to have a higher accumulation rate. However, looking at the localization after 3 h, accumulation in the thyroid gland was very low, confirming that it was transient. In addition, the urinary D-body content exceeded 70% after 3 h, suggesting that the D-isomer was rapidly excreted in the urine (Table 2). A similar study was conducted using tumor-bearing mice. Compared with normal animals, the blood intensity increased 3.9-fold in the D-isomer and 5-fold in the L-isomer. In contrast, accumulation in the thyroid gland and stomach was lower. The accumulation in tumors was 3.25 ± 1.39 %ID/g at 1 h and 3.84 ± 2.10 %ID/g at 3 h for the D-isomer, whereas it was already 7.83 ± 0.30 %ID/g at 1 h and 10.29 ± 0.88 %ID/g at 3 h for the L-isomer, showing that the L-isomer was superior to the D-isomer at all time points. The L-isomer content was higher than that of the D-isomer at all the time points (Table 3). It is well known that the D- and L-isomers of amino acids behave differently throughout the body. For PET probes, the D-isomer is considered desirable because of its low non-specific accumulation [28]. However, the D-isomer is not used as a substrate in vivo [29]. We aimed to develop highly functional compounds for nuclear medicine therapeutics, and while specificity is necessary, a certain amount must be taken up by the tissues. Our results confirmed that the L-isomer was suitable for uptake into tumor tissues using the amino acid therapeutic probe.

^211^At-AAMT-O-Et-L and ^211^At-AAMT-O-Pr-L showed lower cellular uptake when LAT1 expression increased. Because these compounds are more liposoluble than ^211^At-AAMT-O-Me-L, simple diffusion through the lipid bilayer may occur. Therefore, it is thought that compounds that were specifically taken up once may not be retained in the cell (Figure 2A). Although ^211^At-AAMT-O-Me-L had noticeably higher cellular uptake (Figure 2B), this was only an in vitro study, and the results showed that ^211^At-AAMT-O-Me-L had a faster rate of cellular uptake. Retention time in tissues is also relevant to therapeutic effects. Therefore, it may not be correlated with the therapeutic effect in vivo. A comparison of the D- and L-isomers showed similar levels of uptake in the HEK293 cell system (Figure 3A), but more uptake of the L-isomer was observed in the experiment using the cancer cell line PANC-1 (Figure 3B). This suggests that LAT1 expressed in HEK293 cells was artificially expressed and might have been less functional than that expressed in cancer cells. It was also reported that membrane proteins, such as integrin, were complexed with LAT1, and their complexes might reinforce the function of LAT1. Furthermore, the D-isomer amino acid was not utilized in vivo. This might be the reason for the differences in uptake by the cell line.

Next, we examined compounds with a methyl group replaced by an ethyl group in ICR mice, and the accumulation of ^211^At-AAMT-O-Et-L in the thyroid and stomach was very high, and the amount of accumulation was even higher at 3 h than at 1 h. In particular, the accumulation in the stomach at 3 h was very high at 38.74 ± 10.11 %ID/g (Figure 4A). In contrast, in the tumor-bearing model, the accumulation in the stomach was 33.25 ± 8.36 %ID/g at 1 h but this decreased to 18.47 ± 2.59 %ID/g at 3 h (Figure 4B). Both ICR and carrier cancer models showed high accumulation in the spleen and pancreas; the reason for these accumulations might be physiological accumulation, since LAT1 expression is found in the pancreas in rodents more than in humans. The spleen might be a possible physiological accumulation of ^211^At owing to the dehalogenation of the compounds. The accumulation in tumors was 5.64 ± 0.91 %ID/g after 1hr and 5.24 ± 1.39 after 3 h, indicating high tumor retention. We also examined compounds in which the methyl group was replaced by a propyl group. In ICR mice, the accumulation of ^211^At-AAMT-O-Pr-L in the thyroid and stomach was very high, especially at 3 h, and even higher than that at 1 h, at 32.05 ± 3.31 %ID/g in the stomach (Table 5). In contrast, in the carcinoma-bearing model, accumulation in the stomach was 7.31 ± 0.30 %ID/g at 1 h but this increased to 20.60 ± 1.34 %ID/g at 3 h (Figure 4B). The differences in accumulation by the model may be due to strain differences. Histopathological evaluation is necessary to determine whether the differences are due to tumor metastases. The accumulation in tumors was 5.91 ± 0.27 %ID/g at 1 h and 9.45 ± 0.82 %ID/g at 3 h, indicating a higher accumulation in tumors. No significant differences in tumor growth inhibition were observed among the compounds (Figure 4).

Toxicity studies were performed using ^211^At-AAMT-O-Me-L, which appeared to have the highest safety profile based on pharmacokinetic results. Labeled material was administered to ICR mice (male and female) at a dose of 1 MBq/mouse with a clear therapeutic effect, and samples were collected the day after administration and 5, 14, and 28 days later for blood and pathological examination. The results indicated that no effects were observed on the day after administration, and although hematopoietic disorders originating from radiation exposure were observed five days after administration, they were subsequently cured (Appendix A).

In an artificial evaluation system using HEK293 cells, ^211^At-AAMT-O-Me-L was also the best. The amount of uptake was proportional to the expression level of LAT1, the speed of uptake was the fastest, and the short half-life of ^211^At (7.2 h) was advantageous for its rapid uptake property. From the experimental results regarding compound uptake in an artificial system using HEK293 cells, ^211^At-AAMT-OMe-L, which correlates with the expression level of LAT1, was the best. It also exhibited the highest stability. However, some may not consider other compounds inferior. Looking only at the anti-tumor effects, they were almost the same in the PANC1 model, and the amount of uptake may be higher depending on the cancer type. If we focus on tumor-only accumulation rates, both ^211^At-AAMT-O-Et-L and ^211^At-AAMT-O-Pr-L have reasonably good accumulation rates; however, these labeled substances have high accumulation rates in non-specific organs. Although the exact affinity needs to be determined by crystal structure analysis or cryo-electron microscopy, the advantages of ^211^At-AAMT-OMe-L are significant, at least in biological experiments. When considering its use as a drug, there is no doubt that the stability of the labeled compound and the ease of labeling must be considered.

^211^At-AAMT-O-Me-L showed rapid uptake into cells and was very stable. In addition, there was less adsorption on the experimental equipment. Labeling requires only a very small amount of the compound, and the labeling efficiency is high. It also has the simplest structure among candidate compounds. Therefore, we used ^211^At-AAMT-O-Me-L as the hit compound. This compound has been shown to be effective in any cancer that expresses LAT1, regardless of cancer type. The limitations of this study were that the models were limited to one tumor-bearing model for a comparison of the compounds, we did not look at the fine distribution in the time course of all compounds, we did not compare dose-dependent therapeutic effects, and the toxicity evaluation of ^211^At-AAMT-O-Me-L is still in progress. We plan to proceed with clinical applications targeting pancreatic cancer and other tumors with high LAT1 expression levels.

## 4. Materials and Methods

### 4.1. Chemical Synthesis

We presented an optimized design as precursors for astatine labeling using “*Shirakami Reaction*” [14], and asked Kishida Chemical Co., Ltd. (Osaka, Japan) to synthesize them. Water was used as the solvent for the precursor, with boronic acid as a substituent. For the precursor with pinacolborane as the substituent, 7% NaHCO_3_, known as Meylon (Otsuka Pharmaceutical, Tokyo, Japan), was used as the solvent.

### 4.2. Nuclide Manufacturing

Astatine-211 (^211^At) was produced at cyclotron facilities at RIKEN (Wako, Japan) and Research Center for Nuclear Physics at Osaka University (Osaka, Japan). The bismuth deposited on aluminum was irradiated with an α-particle beam. ^211^At was then separated and purified from the irradiated Bi target by dry distillation using the automated dry distillation system, COSMiC-Mini VTRSC2 (Nihon Mediphysics Business Support, Hyogo, Japan), and collected as an aqueous solution [30]. This system was based on previously reported dry separation methods [31].

### 4.3. Chemical Labeling

All the precursors were labeled with ^211^At. We mixed aqueous ^211^At solution with the precursors, added KI as a carrier, and labeled with the “*Shirakami reaction*”. The mixtures were incubated with KI at 50 °C for 50 min. The labeled product was purified using an Oasis HLB column (Waters^TM^, Milford, MA, USA) and its quality was evaluated by high-performance liquid chromatography (HPLC) and thin layer chromatography (TLC).

### 4.4. High-Performance Liquid Chromatography and Thin Layer Chromatography

The radiolabeled products and reference compounds were analyzed by HPLC (LC-20AD, Shimadzu Corp., Kyoto, Japan) using a reversed-phase column (Cosmosil Packed Column 5C_18_ MSII, 4.6 mm × 150 mm, Nacalai tesque, Kyoto, Japan). The samples were eluted using a mixture of Formic Acid and ACN at a flow rate of 1.0 mL/min. The eluate was monitored by a radio-activity flow detector (Gabi star, Elysia-raytest, Angleur, Belgium) and a UV detector (226 nm). Thin-layer chromatography was performed as already described. For thin-layer chromatography (TLC), BAS IP MS 2025E (GE Healthcare Technologies Inc., Chicago, IL, USA) was used for imaging. Image data were collected using a Tyhoon^TM^ FLA-7000 (Cytiva, GE Healthcare Technologies Inc.). A 1:2 mixture of water and acetonitrile was used as the eluent.

### 4.5. Cell Culture

The PANC-1 cell line was obtained from the RIKEN Cell Bank and maintained in RPMI 1640 medium (Fujifilm Wako Pure Chemical, Osaka, Japan) supplemented with 10% inactivated fetal bovine serum (Thermo Fisher Scientific, Waltham, MA, USA). Mock/HEK293, LAT1/HEK293, and LAT2/HEK203 cells were kindly provided by Dr. Kanai [32]. Cell lines controlling LAT1 expression were constructed using the Tet-on system. HEK293 cells were obtained from the RIKEN Cell Bank and maintained in E-MEM (Fujifilm Wako Pure Chemical) supplemented with 10% inactivated fetal bovine serum (Thermo Fisher Scientific) and non-essential amino acids (Fujifilm Wako Pure Chemical). The expression vectors were purchased from VectorBuilder, Inc. (Kanagawa, Japan). The purified viral particle genes were transfected into HEK293 cells. Drug selection and cloning of the cell line (Tet-LAT1/HEK293) were performed. Drug selection was performed using puromycin (Fujifilm Wako Pure Chemical Industries) and hygromycin (Fujifilm Wako Pure Chemical).

### 4.6. Animal Model

#### 4.6.1. Normal Animal Model

ICR mice (male, 5 weeks old) were purchased from Japan SLC (Shizuoka, Japan) and acclimated for one week prior to use. The doses in the distribution experiments were as follows: ^211^At-AAMT-O-Me-L (0.407 ± 0.005 MBq/mouse, 31.64 ± 0.76 g, *n* = 6), ^211^At-AAMT-O-Me-D (0.351 ± 0.004 MBq/mouse, 31.44 ± 0.94 g, *n* = 6), ^211^At-AAMT-O-Et-L (0.685 ± 0.010 MBq/mouse, 24.14 ± 3.31 g, *n* = 6), ^211^At-AAMT-O-Pr-L (1.09 ± 0.06 MBq/mouse, 26.33 ± 0.39 g, *n* = 6)

#### 4.6.2. Xenograft Model

BALB/c nu/nu mice (male, 5 weeks old) were purchased from Japan SLC (Shizuoka, Japan) and subcutaneously transplanted at a density of 1 × 10^7^ cells/mouse. PANC-1 cells that reached the logarithmic growth phase were transplanted subcutaneously by mixing a 1 × 10^8^ cells/mL cell suspension with an equivalent volume of Matrigel (Corning, Corning, NY, USA) to obtain 1 × 10^7^ cells per mouse. Experiments were performed after the size of the transplanted tumor had exceeded 50 mm^3^. The doses in the distribution experiments were as follows: ^211^At-AAMT-OH-L (0.501 ± 0.011 MBq/mouse, 19.91 ± 0.59 g, *n* = 3) and ^211^At-AAMT-O-Me-L (0.384 ± 0.003 MBq/mouse, 19.18 ± 0.38 g, *n* = 6 for Table 3), ^211^At-AAMT-O-Me-L (158.10 ± 2.94 kBq/mouse, 19.27 ± 0.64 g, *n* = 6) and ^211^At-AAMT-O-Me-D (135.08 ± 2.35 kBq/mouse, 18.64 ± 0.97 g, *n* = 6), ^211^At-AAMT-O-Et-L (0.682 ± 0.011 MBq/mouse, 20.25 ± 1.41 g, *n* = 6), ^211^At-AAMT-O-Pr-L (0.927 ± 0.020 MBq/mouse, 21.74 ± 0.37 g, *n* = 6).

In the treatment experiments, the following doses were administered; ^211^At-AAMT-OH-L (1.02 ± 0.02 MBq/mouse, 19.73 ± 0.38 g, *n* = 3) and ^211^At-AAMT-O-Me-L (1.01 ± 0.02 MBq/mouse, 20.23 ± 0.17 g, *n* = 3), ^211^At-AAMT-O-Et- (1.02 ± 0.02 MBq/mouse, 19.63 ± 0.24 g, *n* = 3), ^211^At-AAMT-O-Pr-L (1.02 ± 0.01 MBq/mouse, 18.07 ± 0.24 g, *n* = 3), and control mice (20.33 ± 0.03 g). Observations were conducted three times a week, and tumor size and weight were measured.

### 4.7. Statistical Analysis

The results are expressed as mean ± standard error. Comparisons between the groups were performed using unpaired *t*-tests in Microsoft Excel (version 2016). For multiple comparisons among the three groups, Bonferroni correction was performed. Differences were considered statistically significant at *p* < 0.05.

## 5. Conclusions

There are three advantages of the molecular targeting of LAT1 for the treatment of cancer. First, LAT1 is highly correlated with the degree of cancer progression, and compounds that molecularly target LAT1 are likely to be taken up in large amounts in correlation with LAT1 expression [33,34,35,36,37,38,39]. Thus, compounds targeting LAT1 are selectively incorporated into cancers with a high degree of progression, and high therapeutic efficacy can be expected [40,41,42]. Second, there is a high correlation between LAT1 expression and cancer motility or metastatic potential [43]. In other words, it is possible to treat both primary and metastatic tumors simultaneously. Third, a high expression of LAT1 has been observed in all types of cancer [44,45,46,47,48,49]. This may allow for the treatment of multiple and potential cancers. Second, there are three advantages to using short-lived α-ray nuclear medicinal therapeutic agents. α-Rays have a short range; therefore, there is little radiation exposure to the surrounding environment. Second, nuclear medicine therapeutic agents are injectable, which means that they are less invasive. This means that they are more likely to be used by patients who are not physically strong. Surgery is the mainstay of cancer treatment; however, there is a great need for patients whose quality of life is reduced by resection. Thirdly, ^211^At has a short lifespan. The short lifespan of ^211^At is expected to shorten the hospitalization period among nuclear medicine therapeutic agents [50,51,52]. The constitutional criteria for patients undergoing nuclear medicine treatment were compiled by the ICRP approximately 20 years ago [53,54] but should be revised considering the recent development of new nuclear medicine therapeutic agents.

^211^At-AAMT-OMe-L was found to be the best compound for use as a nuclear medicine therapeutic agent with LAT1 high selectivity. In this study, PANC-1 was used as a tumor-bearing model, but further studies are needed to expand the indications for other cancer types and to investigate safety issues in the future.

## 6. Patents

The results obtained in this study were also applied to patents in Japan and a PCT application (WO/2024/063095).

## Figures and Tables

**Figure 1 ijms-25-12386-f001:**
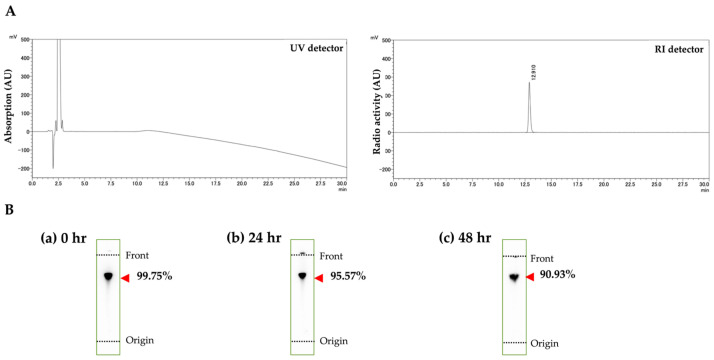
(**A**) HPLC analysis of ^211^At-AAMT-OMe-L. ^211^At-AAMT-OMe-L was purified by an HLB column before HPLC analysis. (**B**) TLC analyses of ^211^At-AAMT-OMe-L. Samples were collected 0, 24, and 48 h after labeling with ^211^At. (**a**) immediately after labeling, (**b**) 24 h after labeling, and (**c**) 48 h after labeling.

**Figure 2 ijms-25-12386-f002:**
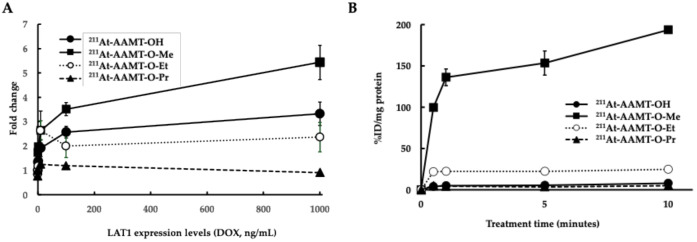
Comparison of uptake between the labeled compounds. (**A**) Relation between uptake ratio and LAT1 expression level. Uptake ratios were compared between LAT1-Tet/HEK293 cells. The Y-axis represents the uptake ratio, and the X-axis is concentration of Doxycycline (Dox) (TaKaRa Bio, Shiga, Japan). The collection time was 30 min after treatment. The expression of LAT1 increased with increasing amounts of Dox. (**B**) Comparison of uptake in PANC-1 cell line. The collection times were 0.5, 1, 5 and 10 min after treatment.

**Figure 3 ijms-25-12386-f003:**
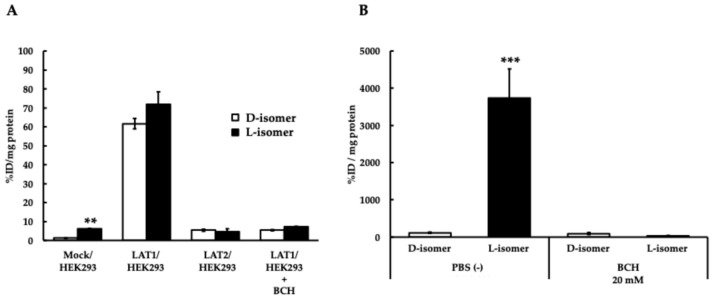
Comparison of intracellular uptake of ^11^At-AAMT-O-Me L-isomer and ^211^At-AAMT-O-Me D-isomer of ^211^At-AAMT-O-Me. (**A**) Uptake in Mock/HEK293, LAT1/HEK293, and LAT1/HEK293 cell lines. (**B**) Uptake in PANC-1 cells. The y-axis represents %ID/mg protein. BCH: -Amino-2-norbornanecarboxylic acid. BCH was treated at 20 mM. ** *p* < 0.01, *** *p* < 0.001.

**Figure 4 ijms-25-12386-f004:**
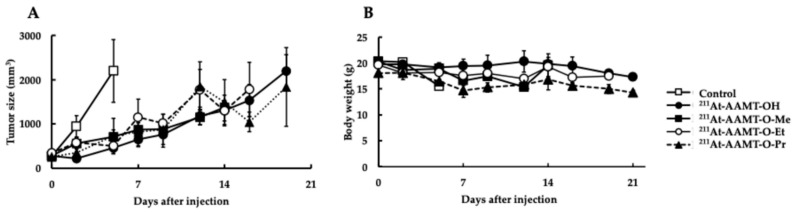
Anti-tumor effects in the PANC-1 xenograft model. (**A**) Tumor sizes. (**B**) Body weights. Each group consisted of three mice. All data are presented as means ± standard errors.

**Table 1 ijms-25-12386-t001:** Comparison of tissue distribution between ^211^At-AAMT-O-Me and ^211^At-AAMT-OH in nude mice (PANC-1 tumor model) 24 h after injection.

	OH	O-Me
**Brain**	0.68 ± 0.16	0.13 ± 0.04
**Thyroid**	11.65 ± 4.36	9.37 ± 3.33
**Salivary glands**	0.42 ± 0.17	13.65 ± 5.31
**Heart**	2.29 ± 1.20	1.29 ± 0.42
**Lung**	1.90 ± 0.55	4.43 ± 1.76
**Liver**	1.02 ± 0.60	0.92 ± 0.15
**Stomach**	1.84 ± 0.81	8.89 ± 2.65
**Small intestine**	0.09 ± 0.01	1.49 ± 0.29
**Colon**	1.03 ± 0.34	1.69 ± 0.15
**Kidney**	0.07 ± 0.02	1.67 ± 0.49
**Pancreas**	2.39 ± 2.03	0.59 ± 0.17
**Spleen**	0.72 ± 0.22	3.12 ± 2.06
**Testis**	0.57 ± 0.44	1.18 ± 0.25
**Blood**	0.04 ± 0.02	0.88 ± 0.23
**Bladder**	1.01 ± 0.10	0.55 ± 0.13
**Bone**	4.29 ± 3.90	0.75 ± 0.18
**Tumor**	0.71 ± 0.46	1.86 ± 0.39
**Cecum**	6.60 ± 2.78	2.09 ± 0.41

All the data are presented as %ID/g (*n* = 3). All data are presented as mean ± standard error. OH; ^211^At-AAMT-OH-L, O-Me; ^211^At-AAMT-O-Me-L. All groups consisted of three mice.

**Table 2 ijms-25-12386-t002:** Comparison of tissue distribution between the D-isomer and L-isomer of ^211^At-AAMT-O-Me in ICR mice.

	1	Hour	3	Hour
	D-Isomer	L-Isomer	D-Isomer	L-Isomer
**Brain**	0.15 ± 0.01	0.32 ± 0.02	0.04 ± 0.00	0.10 ± 0.02
**Thyroid**	8.06 ± 3.56	13.12 ± 1.72	0.42 ± 0.20	0.65 ± 0.16
**Salivary glands**	4.67 ± 1.03	7.16 ± 1.54	0.96 ± 0.16	1.77 ± 0.19
**Heart**	3.20 ± 1.26	2.01 ± 0.15	0.43 ± 0.20	0.34 ± 0.02
**Lung**	2.84 ± 1.11	6.42 ± 1.52	0.69 ± 0.28	1.34 ± 0.32
**Liver**	0.58 ± 0.11	1.11 ± 0.10	1.34 ± 0.47	2.57 ± 0.17
**Stomach**	6.34 ± 2.74	13.65 ± 2.40	8.08 ± 3.50	16.96 ± 1.89
**Small intestine**	2.66 ± 1.23	1.82 ± 0.08	5.33 ± 2.18	3.91 ± 0.20
**Colon**	2.54 ± 1.55	0.93 ± 0.13	0.95 ± 0.59	0.80 ± 0.12
**Kidney**	1.48 ± 0.39	2.47 ± 0.24	0.73 ± 0.15	1.11 ± 0.15
**Pancreas**	2.38 ± 1.36	1.52 ± 0.17	0.42 ± 0.28	0.19 ± 0.04
**Spleen**	2.92 ± 1.49	5.01 ± 1.42	0.42 ± 0.14	0.80 ± 0.19
**Testis**	1.84 ± 0.23	2.31 ± 0.04	0.31 ± 0.02	0.52 ± 0.06
**Blood**	1.90 ± 0.57	1.57 ± 0.10	0.38 ± 0.16	1.17 ± 0.24
**Bladder**	10.62 ± 8.22	7.10 ± 1.15	0.19 ± 0.11	0.10 ± 0.01
**Bone**	0.93 ± 0.35	1.33 ± 0.16	0.28 ± 0.11	0.53 ± 1.31
**Thymus**	1.89 ± 0.90	2.03 ± 0.50	0.18 ± 0.06	0.14 ± 0.03
**Cecum**	0.47 ± 0.19	0.85 ± 0.04	0.50 ± 0.23	0.95 ± 0.11

All the data are presented as %ID/g (*n* = 3). All data are presented as means ± standard errors. All groups consisted of three mice.

**Table 3 ijms-25-12386-t003:** Comparison of tissue distribution between the D-isomer- and L-isomers of ^211^At-AAMT-O-Me-L in nude mice (PANC-1 tumor model).

	1	Hour	3	Hour
	D-Isomer	L-Isomer	D-Isomer	L-Isomer
**Brain**	0.15 ± 0.01	0.46 ± 0.00	0.10 ± 0.03	0.29 ± 0.01
**Thyroid**	0.14 ± 0.04	0.20 ± 0.02	0.14 ± 0.06	0.23 ± 0.04
**Salivary glands**	0.30 ± 0.10	0.58 ± 0.05	0.33 ± 0.13	0.78 ± 0.18
**Heart**	0.53 ± 0.08	0.86 ± 0.04	0.37 ± 0.03	0.76 ± 0.07
**Lung**	2.24 ± 0.89	5.16 ± 0.11	1.27 ± 0.40	5.10 ± 0.42
**Liver**	3.09 ± 0.39	4.92 ± 0.24	2.60 ± 0.07	4.62 ± 0.20
**Stomach**	1.43 ± 0.94	1.32 ± 0.06	1.29 ± 0.77	1.31 ± 0.32
**Small intestine**	2.05 ± 0.81	3.40 ± 0.36	1.48 ± 0.37	2.50 ± 0.42
**Colon**	0.85 ± 0.49	1.14 ± 0.27	0.40 ± 0.25	0.68 ± 0.23
**Kidney**	1.40 ± 0.56	3.56 ± 0.09	1.54 ± 0.86	2.29 ± 0.07
**Pancreas**	0.61 ± 0.44	0.58 ± 0.01	0.48 ± 0.35	0.36 ± 0.10
**Spleen**	1.70 ± 0.74	4.34 ± 0.79	1.81 ± 1.19	3.97 ± 0.59
**Testis**	1.05 ± 0.77	0.55 ± 0.02	1.64 ± 1.26	0.55 ± 0.11
**Blood**	3.52 ± 1.89	7.85 ± 1.17	1.92 ± 1.16	10.38 ± 2.28
**Bladder**	1.76 ± 1.64	0.06 ± 0.00	1.04 ± 0.97	0.09 ± 0.01
**Bone**	0.30 ± 0.12	0.68 ± 1.27	0.26 ± 0.12	0.51 ± 1.35
**Tumor**	3.25 ± 1.39	7.83 ± 0.30	3.84 ± 2.10	10.29 ± 0.88
**Cecum**	2.35 ± 1.73	0.46 ± 0.05	1.32 ± 0.96	0.43 ± 0.16

All the data are presented as %ID/g (*n* = 3). All data are presented as means ± standard errors. All groups consisted of three mice.

**Table 4 ijms-25-12386-t004:** Comparison of tissue distribution of ^211^At-AAMT-O-Et-L in ICR and nude mice (PANC-1 tumor model) 1 and 3 h after injection.

	ICR	Mice		Nude	Mice
	1 h	3 h		1 h	3 h
**Brain**	0.37 ± 0.07	0.13 ± 0.05	**Brain**	0.65 ± 0.11	0.36 ± 0.10
**Thyroid**	8.98 ± 4.12	20.22 ± 6.28	**Thyroid**	24.50 ± 4.40	31.26 ± 11.36
**Salivary glands**	7.08 ± 2.00	15.05 ± 7.81	**Salivary glands**	13.69 ± 2.13	19.70 ± 2.93
**Heart**	6.14 ± 3.75	1.87 ± 0.80	**Heart**	4.41 ± 0.46	5.15 ± 0.95
**Lung**	4.47 ± 1.34	4.93 ± 1.95	**Lung**	9.69 ± 1.20	10.67 ± 1.28
**Liver**	1.41 ± 0.33	1.06 ± 0.21	**Liver**	2.54 ± 0.28	1.93 ± 0.23
**Stomach**	19.50 ± 5.82	38.74 ± 10.11	**Stomach**	33.25 ± 8.36	18.47 ± 2.59
**Small intestine**	12.85 ± 7.35	3.21 ± 0.88	**Small intestine**	4.80 ± 0.63	3.23 ± 0.22
**Colon**	8.32 ± 4.87	1.75 ± 0.49	**Colon**	3.80 ± 0.50	3.63 ± 0.34
**Kidney**	4.70 ± 1.68	1.56 ± 0.13	**Kidney**	6.23 ± 0.94	3.74 ± 0.78
**Pancreas**	7.31 ± 4.45	1.20 ± 0.40	**Pancreas**	4.56 ± 0.82	2.01 ± 0.20
**Spleen**	2.90 ± 0.95	2.09 ± 1.08	**Spleen**	10.32 ± 4.17	13.31 ± 2.43
**Testis**	2.36 ± 0.28	1.20 ± 0.36	**Testis**	3.29 ± 0.07	3.32 ± 0.33
**Blood**	2.03 ± 0.61	0.87 ± 0.27	**Blood**	3.63 ± 0.43	2.23 ± 0.06
**Bladder**	53.22 ± 43.79	21.28 ± 15.54	**Bladder**	15.41 ± 4.39	12.03 ± 3.42
**Bone**	0.71 ± 0.32	1.00 ± 0.18	**Bone**	1.92 ± 0.21	1.41 ± 0.16
**Thymus**	1.14 ± 0.68	0.26 ± 0.17	**Tumor**	5.64 ± 0.91	5.24 ± 1.39
**Cecum**	0.68 ± 0.25	1.67 ± 0.24	**Cecum**	2.36 ± 0.52	2.80 ± 0.56

All the data are presented as %ID/g (*n* = 3). All data are presented as means ± standard errors. All groups consisted of three mice. h; hour.

**Table 5 ijms-25-12386-t005:** Comparison of tissue distribution of ^211^At-AAMT-O-Pr-L. A. Distribution in ICR at 1 h and 3 h after injection. B. Distribution in nude mice (PANC-1 tumor model) at 10 min and 1 h. All the data are presented as %ID/g.

	ICR	Mice		Nude	Mice
	1 h	3 h		10 min	1 h
**Brain**	0.25 ± 0.02	0.16 ± 0.01	**Brain**	1.01 ± 0.04	0.79 ± 0.13
**Thyroid**	23.08 ± 3.56	24.75 ± 3.77	**Thyroid**	9.74 ± 0.39	16.44 ± 1.16
**Salivary glands**	14.23 ± 3.16	11.39 ± 0.33	**Salivary glands**	11.89 ± 0.06	20.98 ± 3.26
**Heart**	3.14 ± 0.20	2.16 ± 0.18	**Heart**	9.66 ± 0.87	5.39 ± 1.06
**Lung**	6.54 ± 0.34	5.09 ± 0.43	**Lung**	18.16 ± 1.88	14.15 ± 1.78
**Liver**	1.18 ± 0.08	0.82 ± 0.14	**Liver**	3.67 ± 0.41	2.41 ± 0.24
**Stomach**	13.55 ± 2.66	32.05 ± 3.31	**Stomach**	7.31 ± 0.30	20.60 ± 1.34
**Small intestine**	1.83 ± 0.23	1.92 ± 0.12	**Small intestine**	6.80 ± 0.63	5.08 ± 0.53
**Colon**	1.35 ± 0.07	1.29 ± 0.25	**Colon**	4.56 ± 0.31	3.27 ± 0.46
**Kidney**	2.25 ± 0.19	1.93 ± 0.29	**Kidney**	10.29 ± 0.97	5.76 ± 0.67
**Pancreas**	2.55 ± 0.60	1.60 ± 0.31	**Pancreas**	10.85 ± 1.53	3.54 ± 0.49
**Spleen**	4.11 ± 0.11	3.56 ± 0.54	**Spleen**	11.84 ± 1.91	17.42 ± 6.48
**Testis**	3.66 ± 0.14	2.34 ± 0.42	**Testis**	3.54 ± 0.28	3.56 ± 0.41
**Blood**	1.77 ± 0.15	1.30 ± 0.12	**Blood**	4.33 ± 0.45	3.09 ± 0.25
**Bladder**	20.82 ± 5.36	5.37 ± 0.73	**Bladder**	18.24 ± 4.50	18.00 ± 0.97
**Bone**	1.79 ± 0.12	1.32 ± 0.12	**Bone**	3.50 ± 0.13	2.03 ± 0.45
**Thymus**	2.03 ± 0.28	1.76 ± 0.25	**Tumor**	5.91 ± 0.27	9.45 ± 0.82
**Cecum**	1.49 ± 0.14	1.13 ± 0.25	**Cecum**	3.35 ± 0.30	3.07 ± 1.06

All the data are presented as %ID/g (*n* = 3). All data are presented as means ± standard errors. All groups consisted of three mice. h; hour.

## Data Availability

Can be found within the article.

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
