# Peer review of "Development of LAT1-Selective Nuclear Medicine Therapeutics Using Astatine-211"

_ijms, 2024, doi:10.3390/ijms252212386_

Round 1
Reviewer 1 Report
Comments and Suggestions for Authors
This paper describes a complete study of new small molecules radiolabeled with astatine-211 targeting L-type amino acid transporter 1 (LAT1), a cancer cell-type transporter. More precisely, this study focused on the preparation as well as in vitro and in vivo evaluation of 211At-α methyl-L-tyrosine (211At-AAMT-OH-L) derivatives. In the literature, 211At-AAMT-OH-L was already reported by the authors as promising therapeutic agent in pancreatic cancer model. The idea of the work detailed here is to go further with the investigation of 211At-AAMT structurally modified with different substituent at the 4-position of the aromatic ring (methyl, ethyl or propyl group). For that purpose, four 211At-AAMT derivatives were prepared (211At-AAMT-O-Me-L, 211At-AAMT-O-Me-D, 211At-AAMT-O-Et and 211At-AAMT-O-Pr) from the corresponding arylboronic precursors and using a radiolabeling procedure already described. These four astatinated compounds were obtained with interesting RCY (75-81%) and high RCP (98-99%). Comparison of intracellular uptake following LAT1 expression levels or treatment time revealed that 211At-AAMT-O-Me-L exhibited the highest and the fastest accumulation among all other derivatives. The specificity of LAT1 was also test with L- and D- isomers demonstrating that L- isomer was specific for PANC-1 cells when both isomers were interesting with LAT1 modified HEK293 cells. Biodistribution studies of 211At-AAMT-O-Me-L showed a reduced accumulation in bone and pancreas, a higher uptake in tumor but also in salivary glands, lungs, spleen and stomach compared to 211At-AAMT-OH-L. Other tissue distribution experiments highlighted a relative similar profile between L- and D- isomers of 211At-AAMT-O-Me, with a high uptake in thyroid, stomach or salivary glands in ICR mice. In nude mice with PANC-1 tumor model, distribution was comparable for both compounds with a low uptake in non-target organs compared to ICR mice, but especially a high tumor accumulation for L- isomer. Concerning distribution profile of 211At-AAMT-O-Et and 211At-AAMT-O-Pr in the two models tested (ICR mice and Nude mice with PANC-1 cells), both compounds showed a very high uptake in target organs of free astatine (thyroid, stomach, lungs, salivary glands), increasing the potential risk of side effects. Unfortunately, therapeutic study in the PANC-1 tumor model showed no difference between all the four astatinated compounds (211At-AAMT-OH-L, 211At-AAMT-O-Me-L, 211At-AAMT-O-Et and 211At-AAMT-O-Pr).
The manuscript is comprehensive, structured clearly and logically, furthermore, the work presented is significant. The general strategy is easy to follow, results are carefully reported and the discussion is consistent and detailed. The work described here is in the continuity of previous studies reported by the authors on the development of 211At-α methyl-L-tyrosine (211At-AAMT-OH-L) as therapeutic agent. Even if the originality is quite limited, the novelty brought by investigations and results obtained with 211At-AAMT derivatives is interesting and allow to accumulate data for the development of astatinated small molecules. However, the work presented raised some questions or comments that will be detailed below and that can be considered as ideas for improvement of the general discussion. Finally, in view of the work carried out, the results obtained and despite the few questions/comments noticed, I recommend a publication of this article after minor revision.

Author Response
Dear Reviewer 1
Thank you very much for taking the time to review this manuscript. Please find the detailed responses below and the corresponding revisions/corrections highlighted/in track changes in the re-submitted files.
Point-by-point response to Comments:
Comments 1: L22 : For greater clarity, it is not necessary to mention the labeling method here
Response 1: Thank you for pointing this out. We changed “Shirakami reaction” to “a unique method”.
Comments 2: L42 : 8 references to illustrate applications of 211At is maybe too much, it could be better to select the most striking.
Response 2: Labeling using astatine has been attempted with a variety of small molecules, medium molecules, and polymers. We label small molecules. We have cited many examples to demonstrate the wide range of applications of astatine.
Comments 3: 2.1. Synthesis of 211At labeled chemicals
L85-86 : “Compounds labeled using the mercury method were unstable”, does that mean
that the astatinated compound was not stable or that the mercury intermediate involved in
the reaction was not stable ? The final astatinated compound is independent of the labeling
way, in reference 12, it seems that addition of ascorbic acid allows to improve the stability
211At-AAMT. Why compounds would be more stable using the Shirakami reaction? Besides,
RCY described for mercury method are in the same range than those obtained in this paper
with the Shirakami reaction. It could be interesting to give more details to clarify this section.
Response 3: Yes. Separation by mercury method was performed using a cation exchange column and an anion exchange column in tandem. Labeling requires nearly three hours, so it is possible that the astatine has oxidized by the time the labeling is complete, or that the trace amount of mercury mixed in has some effect to stability of astatinated compound. In the Shirakami reaction, an exchange group with astatine has already been inserted into the precursor. We speculate that this restricts the sites where astatine reacts, making astatinated compound more stable. Almost no decomposition occurs even without ascorbic acid (it is often added as it prevents adsorption to the tube). Furthermore, almost no unnecessary reagents are used as carriers during the reaction, and the reaction proceeds under near-neutral conditions. It is thought that these factors also contribute to the improvement of stability. These things are still in speculation. We are proceeding with the verification of these phenomena to further strengthen the usefulness of Shirakami Reaction. By collecting and verifying data, we would be able to prove our hypothesis.
Comments 4: Did you prepare the corresponding non-radioactive iodinated reference of the labeled compounds in order to validate the retention time and thus to validate the characterization? As illustration, a chromatogram with iodinated reference and astatinated compounds would be interesting.
Response 4: The Shirakami reaction can also be used to label iodine. However, the amount of precursor was too small to be detected using stable isotopes, so verification using iodine has not been possible. We tested another compound with iodine by HPLC, but there was almost no difference between iodinated and astatinated compound.
Comments 5: Is the concentration of the precursor solution used for the labeling was as 1mg/mL as reported in reference 20?
Response 5: Yes, it is. Concentration of compound is related with intensity of radiation. According to amount of 211At, we could change the amount of precursor.
Comments 6: Is it certain that the precursor is eliminated after purification using Oasis HLB column in the case of 211At-AAMT derivatives? Did you estimate the specific activity of final astatinated compounds?
Response 6: Yes. The amounts of substances labeled with radioactive isotopes is so small that they can hardly be detected with a UV detector. By comparing the precursor before the reaction and looking at the data from the UV and radiation detectors of astatinated compound, it can be shown that only the radioactively labeled compounds are present.
Comments 7: Concerning radiolabeling conditions, in the reference 20, labeling was performed at room temperature during 30 min, when this described here was at 50°C during 50 min. Why this difference in labeling conditions? Did you try to optimize it?
Response 7: As a result of optimization, 50°C was optimal for this series of compounds. Although the reaction itself proceeded at room temperature (25°C), it took a long time. In addition, although the reaction progresses faster (20 min) at high temperatures (95°C), there were problems with the stability of the compound. Temperature and time must be optimized for each compound.
Comments 8: Did you verify with chiral column that labeling conditions did not reverse isomer configuration when labeling L- or D- isomer?
Response 8: D- and L-isomer precursors were prepared separately. The chiral column was used in the production of the precursor, but the compound was not analyzed in the chiral column after labeling. Radiolabeled compounds were difficult to separate on the column because of the low amount of substance.
Comments 9: L117 (Figure 1A) : Specify if analysis are before or after purification.
Response 9: Thank you for pointing that out. We wrote the information in the figure legend.
Comments 10: 2.3. Distribution differences between candidate compounds in vivo
Generally in all the Tables : Is it bladder instead of bradder ?
Response 10: Thank you for pointing that out. It's an elementary typo. I will correct it “Bladder” in all the Tables.
Comments 11: Uptake in target organs of free astatine is quite high, did you try to compare the distribution using blocking agent?
Response 11: Yes, that an ongoing experiment. In vitro study, we can inhibit the uptake of compound. However, DNA damage, colony formation, and viability are not inhibited by blocking agent in vitro study. It is because in monolayer culture, thick of cell is around 1-2μm, thus α-ray could reach to the nucleus of the cell from outside.
LAT1 inhibitors alone have anti-tumor effects. 211At-AAMT-OMe-L and LAT1 inhibitors also have short biological half-lives. Therefore, we are currently investigating the timing of LAT1 inhibitor and 211At-AAMT-OMe-L administration, and the time point at which the inhibitory effect is most visible.
Comments 12: Table 2 vs. Table 3: How do you explain the high uptake in target organs of free astatine in ICR mice and not in nude mice with PANC-1 tumor model?
Response 12: Compounds with high LAT1 selectivity are selectively taken up by organs with higher LAT1 expression. For example, amino acid derivatives pass through the blood-brain barrier, but if there is a tissue with significantly high levels of LAT1, such as a brain tumor, they will be taken up by the highly expressing tissue rather than by the surrounding normal tissue.
An important point revealed by these data is cancer-specificity and selectivity of labeled compounds. In the tissues that been labeled compounds taken up in normal animals no longer accumulate in the same tissues of tumor-bearing animals. This means that it is highly safe for cancer patients. In addition, even in cases where some amount of uptake has occurred in normal animals, we have confirmed that there is no histological effect or that the level is curable. In other words, it can be said that even if uptake is confirmed over a short period of time, if it is quickly excreted, there is no obvious toxicity to the individual.
Comments 13: T L207-211: The section “In nude mice…same manner as 211At-AAMT-O-Et-L” : This part is not clear, please reword the sentences.
Response 13: Thank you for pointing that out. We change the sentences and changes were highlighted.
“Data from ICR and nude mice for labeled compounds in which the methyl group of the side chain was changed to an ethyl group, with the aim of increasing accumulation in tumors, are shown in Table 4. In ICR mice, the accumulation of 211At-AAMT-O-Et-L in non-specific organs was very high, and side effects were feared (Table 4A). In Nude mice (PANC-1 tumor-bearing model), tumor accumulation and non-specific accumulation in non-tumor organs were higher (Table 4B).”
Changed to
“We changed the side chain the methyl group (211At-AAMT-OMe-L) to an ethyl group (211At-AAMT-OEt-L) with the aim of increasing accumulation in tumors. The distribution data in ICR mice or nude mice (PANC-1 tumor-bearing model) were shown in Table 4. In ICR mice, the accumulation of 211At-AAMT-O-Et-L in non-specific organs was very high, and side effects were feared (Table 4A). In Nude mice, tumor accumulation and non-specific accumulation in non-tumor organs were also higher (Table 4B).”
Comments 14: 2.5. Anti-tumor effect in xenograft model
How do you explain the moderate therapeutic effect of the compounds and especially 211At-AAMT-O-Me-L that showed the highest tumor uptake?
Response 14: This review used the same cancer type (PANC1) for comparison with our previous report (Cancer Sci. 2021). However, subsequent studies have identified several more effective cancer types. In other words, two reasons may explain this discrepancy between efficacy and uptake. First, there is the issue of cancer type with respect to therapeutic efficacy. We believe that compound differences would be reflected to a greater extent if cancer types that are more sensitive to radiation were used. Second, we believe that the speed of excretion of the compounds may be a factor. If safety is taken into consideration, rapid excretion would be appreciated. Therefore, we are considering the possibility that multiple dosing may be more suitable for treatment with this compound.
Comments 15: L279 : Totally agree for this tendency, but did you determine the log P value in order to confirm it?
Response 15: Thanks for your advice. I have only synthesized in mg order because the amount used in nuclear medicine therapeutics is very small, thus the amount of compound we synthesized was too small. I will synthesize additional compounds as soon as possible to obtain the Log P, although I will not be able to meet the review deadline.
Comments 16: Supplementary Please indicate if TLC refers to crude reaction or compounds after purification, addition of the corresponding Rf could be also interesting.
Response 16: Thank you for pointing this out. These data were compounds after purification. We added the information in the figure legend (Supplementary Figure 2).
Thank you for your kind and meaningful comment. We have responded to your comments to the best of our knowledge. The revisions are also highlighted in the text. Thank you in advance.
Kazuko Kaneda-Nakashima, Ph.D.
MS-CORE, FRC, Osaka University Graduate School of Science
Reviewer 2 Report
Comments and Suggestions for Authors
Title: Development of LAT1-selective nuclear medicine therapeutics using astatine-211
My comments:
1. Figure 1: Can authors add the buffer system to the iTLC figure description? What was the buffer system used for the iTLC?
Figure 1A: Must authors include the UV detector? It shows nothing. I suggest authors take it off and probably include the blank run instead if you like.
2. After substituting -OH with -CH3 group did authors determine octanol-water partition ratio? This will be interesting as it will support your claim.
3. What was the explanation for the observation at paragraph 85-88?
4. Figure 3 is not clear to me. Did you perform any blocking studies? If no, then it has to be done. If yes, then please, label it properly.
5.Will there be a stable supply for 211At should this radiotracer enter the clinic?
Author Response
Dear Reviewer 2,
Thank you very much for taking the time to review this manuscript. Please find the detailed responses below and the corresponding revisions/corrections highlighted/in track changes in the re-submitted files.
Point-by-point response to Comments:
Comments 1: Figure 1: Can authors add the buffer system to the iTLC figure description? What was the buffer system used for the iTLC?
Response 1: Thank you for pointing this out. TLC was performed using a 1:2 mixture of water and acetonitrile as the development solvent, and signals were acquired on an imaging plate purchased from GE, and signals acquired on a FLA7000 were detected as data. We have added a note to that effect in the text.
Comments 2: Figure 1A: Must authors include the UV detector? It shows nothing. I suggest authors take it off and probably include the blank run instead if you like.
Response 2: Thank you for pointing this out. It indicates that the labeled compound itself is a trace amount. In other words, it indicates that the amount is detectable with a radiation detector but is difficult to detect as a normal amount of substance. This may also indicate that the toxicity of the unlabeled compound is negligible due to the small amount of the labeled compound.
Comments 3: After substituting -OH with -CH3 group did authors determine octanol-water partition ratio? This will be interesting as it will support your claim.
Response 3: Thank you for pointing this out. Certainly, using that parameter would be a good indicator to evaluate accommodability. After we synthesized the compounds, we ran them through HPLC and mass spectrometry to see if they were manufactured as designed, but we did not conduct this due to the small number of compounds we synthesized. Thank you for your advice, we will try to do that analysis. We will try to implement it as soon as possible, although we will not meet this deadline.
Comments 4: What was the explanation for the observation at paragraph 85-88?
Response 4: Thank you for pointing this out. We are reporting in 2021 using the mercury method. This one show that the labeled compound would decompose in a few hours without an antioxidant. However, as you can see in Figure 1B of this article, the compound produced using the Shirakami reaction was very stable. We will add a note to that effect in this section.
“Compounds labeled using the mercury method were unstable without the addition of an antioxidant (ascorbic acid) [12]; however, compounds created by the “Shirakami reaction” were very stable. The data for (S)-2-amino-3-(3-(astato-211At)-4-methoxyphenyl)-2-methylpropanic acid (211At-AAMT-OMe-L) as a representative compound are shown.”
Changed following and inserted to after Line 98
“Compounds labeled using the mercury method were unstable without the addition of an antioxidant (ascorbic acid) [12]; on the other hand, the data for 211At-AAMT-OMe-L shown in Figure 1B shows that the compound made by the Shirakami reaction is stable.”
Comments 5: Figure 3 is not clear to me. Did you perform any blocking studies? If no, then it has to be done. If yes, then please, label it properly.
Response 5: Figure 3 shows a cell line in which LAT1 or LAT2 was forced into cells using pcDNA3.1 based on HEK293 cells, which are not cancer cells and originally showed little amino acid uptake. Mock/HEK293 is a cell line in which pcDNA3.1, the empty vector of the vector used for transduction, was introduced. It does not differ from the parental strain except that it has acquired drug resistance. BCH is an inhibitor of the LAT family and inhibits the function of LAT1, 2, 3, and 4. There was no significant difference between the D-isomer and the L-isomer in the artificial experiment.
Figure 3B is an examination in cancer cells. Since cancer cells originally express LAT1, there is no equivalent to such as Mock/HEK293 and LAT2/HEK293 in the HEK293 experimental system. Therefore, the two graphs are with and without inhibitors. We also tried to inhibit LAT1 using shRNA vectors and other methods, but since LAT1 expression is essential for cancer cell survival, the knockdown strain could not survive and could not be established as a strain. We believe that we were able to show that LAT1-specific uptake was suppressed by inhibiting LAT1 function with BCH.
Comments 6: Will there be a stable supply for 211At should this radiotracer enter the clinic?
Response 6: Thank you for your comments. 211At is made by our AVF cyclotron. This accelerator is a bit large and would be difficult to place in the clinic. 211At will deliver from manufacturing base. As a test of delivery, we transported the material made in our accelerator from Saitama to Osaka (534.6 km) and confirmed that it can be separated and labeled without problems at the place where we received it. 211At solution separated in Osaka was transported from Osaka to Kanazawa (308 km) to see if it could be used without any problem. We also confirmed that the labeled compound could be delivered from Osaka to Kyoto (56.4 km) without any problem.
For the use of 211At, we believe that if the manufacturing site is within 500 km, there should be no problem if transported. The automatic separation and labeling device are being developed by us. We are already able to perform separation and labeling in hospitals using a prototype machine at our university hospital, and we are using it for clinical trials.
Many thanks for your many kind and meaningful comments. We have answered your comments to the best of our ability. The revisions are also highlighted in the text. Thank you in advance.
Kazuko Kaneda-Nakashima, Ph.D.
MS-CORE, FRC, Osaka University Graduate School of Science